# An initiative to develop capability-adjusted life years in Sweden (CALY-SWE): Selecting capabilities with a Delphi panel and developing the questionnaire

Kaspar Walter Meili[1]*, Anna Månsdotter[1,2], Linda Richter Sundberg[1], Jan Hjelte[3], Lars Lindholm[1]

1 Department of Epidemiology and Global Health, Umeå University, Umeå, Sweden, 2 Department of Living Conditions and Lifestyles, Public Health Agency Public Health Agency of Sweden, Stockholm, Sweden, 3 Department of Social Work, Umeå University, Umeå, Sweden

* kaspar.meili@umu.se

## Abstract

### Introduction

Capability-adjusted life years Sweden (CALY-SWE) are a new Swedish questionnaire-based measure for quality of life based on the capability approach. CALY-SWE are targeted towards use in cost-effectiveness evaluations of social welfare consequences. Here, we first motivate the measure both from a theoretical and from a Swedish policy-making perspective. Then, we outline the core principles of the measure, namely the relation to the capability approach, embedded equity considerations inspired by the fair-innings approach, and the bases for which capabilities should be considered. The aims were to 1) the most vital capabilities for individuals in Sweden, 2) to define a sufficient level of each identified capability to lead a flourishing life, and to 3) develop a complete questionnaire for the measurement of the identified capabilities.

### Material and methods

For the selection of capabilities, we used a Delphi process with Swedish civil society representants. To inform the questionnaire development, we conducted a web survey in three versions, with each Swedish 500 participants, to assess the distribution of capabilities that resulted from the Delphi process in the Swedish population. Each version was formulated with different strictness so that less strict wordings of a capability level would apply to a larger share of participants. All versions also included questions on inequality aversion regarding financial, educational, and health capabilities.

### Results

The Delphi process resulted in the following six capabilities: Financial situation & housing, health, social relations, occupations, security, and political & civil rights. We formulated the

**Data Availability Statement:** The data is publicly available from the Swedish National Data service (SND-ID: 2021-39), under the following three

DOIs: Version A: 10.5878/gbyb-1s51 (https://doi.org/10.5878/gbyb-1s51) Version B: 10.5878/dp5t-rp64 (https://doi.org/10.5878/dp5t-rp64) Version C: 10.5878/t7fa-5w15 (https://doi.org/10.5878/t7fa-5w15).

**Funding:** The study was funded by Forte (Swedish research council for health, working life and welfare) Grant No 2018-00143, principal investigator Lars Lindholm, https://forte.se/. The founders had no role in study design, data collection and analysis, decision to publish, or preparation of the manuscript.

**Competing interests:** The authors have declared that no competing interests exist.

final phrasing for the questionnaire based on normative reasons and the distribution of capabilities in the population while taking into account inequality aversion.

## Conclusion

We developed a capability-based model for cost effectiveness economic evaluations of broader social consequences, specific to the Swedish context.

## Introduction

### Overview

The aim of this paper is to describe a Swedish initiative to develop a new outcome measure for public health and social interventions, capability-adjusted life years (CALY-SWE), for use in economic evaluations [1]. The measure is based on Sen's capability approach [2, 3], and the quality-adjusted life years (QALYs) methodology that is frequently used for economic evaluations in healthcare [4].

The paper is organized as follows. The background section motivates CALY-SWE from a Swedish policy perspective and outlines the historical development in health economics from utilitarianism and cost-benefit analysis to extra-welfarism and QALYs, and how this historic context relates to CALYs.

Then, the principles of our measure are described, followed by two investigations. The first investigation was a Delphi panel with participants from the Swedish civil society, the purpose being to select the capabilities to include in the measure. The second investigation was a survey with Swedish participants in which we investigated the present distribution of capabilities in the Swedish population (n = 1,500) to inform the phrasing of the instrument. We aimed to formulate questions that measure each capability on three levels, where the highest level corresponds to having the opportunity of living a flourishing life.

In the next section, the results from the two empirical investigations are merged, and we suggest a complete questionnaire, including phrasing. Finally, we discuss our proposal for a new outcome measure in relation to other attempts to develop capability-based measures.

### Reasons for a capability-based wellbeing measure in Sweden

QALYs were established in the 1980s and have now spread globally as the primary outcome measure in economic evaluations in healthcare and public health. QALYs combine health-related quality of life and time into a single metric [4].

However, finding appropriate measures for policymakers to evaluate interventions with broader social consequences than health is challenging. In the Swedish context, the public sector has three tiers: national, regional, and municipal. National actors and regions are mainly responsible for healthcare and public health and tend to rely on QALYs, while municipalities are mostly responsible for welfare—for example, primary and secondary education, social care and elderly care. Because municipalities are involved with interventions with consequences primarily beyond population health, QALYs are a mostly useless currency, as they do not directly capture effects from relevant welfare areas.

Furthermore, the theory of social determinants of health is important in public health. However, there is a tendency in public health to treat social living conditions mainly as a means to an end—better health. Mackenbach [5] judges the UK strategy from 1999 to be the most ambitious attempt in its history to reduce health inequalities. Minimum wages were

increased, along with benefits and pensions. Spending on education, housing and urban regeneration was increased. From a public health point of view, the strategy failed, since the health inequalities persisted, which has also been the legacy. However, from a wellbeing perspective, the gains could perhaps have been worthwhile despite the costs. As far as we know, this broader picture has not yet been investigated.

To capture broader social consequences, policy actors could in theory rely on cost-benefit analysis, applying willingness to pay. However, willingness-to-pay studies are not common in the evaluation of public health and social policies. Instead, in Sweden, a simple cost-saving approach is frequently used [6] which can hardly capture the complex social consequences of the interventions under consideration, such as long-term benefits of improved education Costs and savings (that is, reduced costs), on the other hand, can be captured with a high degree of certainty. Therefore, these cost-saving analyses may lead to a bias towards resource-saving interventions, while interventions creating quality of life gains are given low priority.

## From welfarism to extra-welfarism and capabilities

Welfarism has its roots in utilitarianism, developed by Bentham (1748–1833) and Mill (1806–1873) [7]. There are two pillars of utilitarianism: to strive for maximisation of utility or welfare, and actions should be assessed according to their consequences. We need to check whether a policy has good or bad consequences before we assess its value.

These ideas have remained at the core of twentieth-century economics. In neoclassical welfare economics, it is assumed that individuals strive to maximize their utility, which is derived from goods and services consumed. Furthermore, individuals choose rationally, which means that they are able to consistently rank all choices at hand. Another important normative assumption is individual sovereignty—i.e. the individual is the only judge of his/her own welfare [8].

The conceptual origin of cost-benefit analyses stems from the tradition of welfarism. In principle, individuals are asked to evaluate any change in monetary terms. If the consequences are positive (winners), they are asked about the largest amount they are willing to pay to achieve this change. If the consequences are negative (losers), they are asked about the minimum compensation they need to accept the change. If the total amount winners are hypothetically willing to pay is enough to compensate the losers, the net is positive and the total utility in the society will increase. This is known as the potential Pareto criterion or Kaldor-Hicks criterion [9]. Comparisons between individuals are only possible by accepting potential Pareto improvement in the form of compensatory payments according to the Kaldor-Hicks criterion.

Sen opposed welfarism and cost-benefit analysis, such as the behavioural assumptions of rational utility maximization. He voiced his critique, for example, in his 'Rational Fools' paper from 1977 [10], and he went on to develop the capability approach. Specifically, Sen proposed that the most important information to consider is capabilities [2]—i.e. whether an intervention increases individuals' opportunities that they value for living a flourishing life.

Capabilities denote the set of what individuals can be and what they can do. The individuals themselves should have freedom in how to shape their lives by choosing what capabilities they would like to realize, as opposed to measuring wellbeing based on individuals' endowments or utility. Sen writes that capability is the freedom to achieve valuable functionings [11], and illustrates this by the difference between being hungry due to fasting versus starving. A contemporary example from Sweden would be the difference between voluntary retirement and forced retirement.

In parallel to, and influenced by, the development of the capability approach, extra-welfarism was established in health economics. Extra-welfarism extends the information space

beyond individual utility and allows for interpersonal comparisons. QALYs, for example, are based on extra-welfarist values, in that health is the underlying value to be maximized [9].

Brouwer et al. [12] made a comparison between welfarism and extra-welfarism, concluding that the latter accepts, and even strives for, broader outcomes and more collective decision-making. Furthermore, according to the extra-welfarist framework, individuals themselves, as well as experts or the general public, can value the outcomes; explicit equity weights can be added, and interpersonal comparisons are desirable.

The most common interpretation of this extra-welfarist framework is that healthcare interventions should be judged by their impact on health, preferably in terms of QALYs. According to Cookson [13], the QALY measure displays clear characteristics of the capability approach, such as the focus on instruments with multiple attributes that correspond to functionings and approval of non-utilitarian values. For example, estimating QALYs by the well-established instrument EQ-5D implies considering dimensions of mobility, self-care, usual activities, pain/discomfort and anxiety/depression [14].

Capabilities, as an outcome measure in healthcare, elderly care or social services, have gained increasing attention during the last decade [1, 15–18]. While capability-adjusted life years have been suggested and implemented before [19–22], our approach includes equity considerations from the start and is focused on an amalgamation of the QALY approach and a set of capabilities covering vital aspects of a good life relevant for the Swedish policy context.

### Aims

As discussed, there is a need for a new broader measure, and we argue that the QALY approach merged with Sen's capability approach would be suitable. Thus, the aims of this article are:

1. To identify the most vital capabilities for individuals in Sweden.

2. To define a sufficient level of each identified capability for leading a flourishing life.

3. To develop a complete questionnaire for the measurement of the identified capabilities.

## Capability-adjusted life years (CALYs): Our proposal in principle

With the CALY-SWE measure, we propose to transfer the QALY concept of a weighted lifetime to a capability basis. QALY has shown to be very useful in decision-making all over the world, and our intension is to promote CALYs for decision-making guided by both efficiency and equity goals. Whereas CALY-SWE specifically denotes our proposed instrument for CALYs, CALY denotes the capability-weighted life year concept in general, to the distinction between for example EQ-5D and QALYs.

CALY-SWE weights are based on a descriptive system consisting of capabilities which can be satisfied on three levels. The levels of each capability are ordinal and define the extent of a capability set. For each possible configuration of capabilities and levels, a weight between 0 and 1 is calculated that corresponds to an index for the overall quality of life in term of capabilities.

The conceptual anchoring of 1 and 0 and corresponding justifications constitutes another issue. We aim to anchor 1 to a capability set sufficient for living a life that is "flourishing" [23]. The concept of human flourishing goes back to Aristotle and has also been adopted by Martha Nussbaum [24–26].

Similar to QALYs, the weight 0 is given to a capability configuration equivalent to 0 lifetime. QALYs allow negative weights—that is, states judged as being worse than death [27, 28]. However, in this phase of development, we limit the scale to 1 to 0.

The normative assumption in mainstream health economics is the maximization of QALYs subject to budget constraints, following the utilitarian tradition. However, other normative assumptions have been proposed, and they are usually gathered under the label 'trade-offs' between efficiency and equity [29–31]. One version of this idea, known as "fair innings", was suggested by Alan Williams [31]. Williams' point of departure was lifetime QALYs for the UK population according to social position. His data showed a well-known pattern: a low position was associated with relatively few QALYs and a high position with many. Williams suggested that a 'fair innings' (e.g. average lifetime QALYs) should guide resource allocation [31]: those above should get less and those below should get more. He writes:"It [the fair innings concept] reflects the feeling that everyone is entitled to some 'normal' span of health and anyone failing to achieve this has been cheated, whilst anyone getting more than this is 'living on borrowed time'" [31].

A version of this idea, known as prioritarianism, also appears in contemporary philosophy. Prioritarianism holds that an increment in wellbeing is morally more valuable the lower (in absolute terms) the level of wellbeing from which this increment arises [32]. Wellbeing is summed over all individuals, but extra weight is given to the wellbeing of individuals who are worse-off.

Interpretations of prioritarianism dominate the ethics in healthcare and public health. Severity of the condition is perhaps the most important criterion when ascribing priorities in healthcare. Norheim et al. suggest the principles of "greatest benefit" and "worst-off" [33].

Our intention is to follow in the footsteps of Alan Williams and other proponents of prioritarianism, particularly considering the interests of the worst-off. For each capability, we aim to define the sufficient capability level for a flourishing life. For this, we conducted a survey to measure the distribution of capabilities in the Swedish population, reported in the section 'Setting sufficient capability thresholds in Sweden'.

Some capability aspects can easily be measured on continuous scales—e.g. wealth in dollars and health in lifetime QALYs. If sufficient wealth for a flourishing life is decided to be 50,000 USD annually, any income above that threshold does not increase capabilities nor quantity of CALYs. In contrast, personal wellbeing, in terms of utility, and total societal welfare across all individuals may certainly increase when income grows if we undertake a utilitarian calculus. Thus, in the CALY calculation, only improvements below the threshold are counted.

A crucial difference between wealth and health in Nordic/European welfare states is the rights the citizens are given. In the distribution of health care, need is the most important criterion, and even the poor and sick are principally offered the best possible care, which is clearly stated in the Swedish Health Care Law [34]. Most political parties in Europe support healthcare allocation purely according to need and disregard any association between health outcomes and social position.

Regarding wealth, the normative thinking is different. The cash benefits people can get in case of no other incomes are on a low level. Income and wealth distributions are skewed to the right and have a very long tail [35, 36]. Deservedness plays a big role when society assess the fairness of incomes, but less so in the fairness of health outcomes.

Societal ambition is high when it comes to a health capability. The ideal distribution is a long and healthy life for all. To achieve this, we certainly need to allocate more resources to those with the poorest health, along the lines suggested by Williams.

When it comes to wealth, the societal ambition for redistribution is low, or perhaps moderate. The ideal distribution may be quite scattered. Societies usually do not deny people earning millions or even billions of dollars, but financial success should be a private endeavour.

Other capabilities, at least ideally, are dichotomous. All should have equal and complete political rights, and no one deserves to only get partial rights. However, in real life there exist

levels between all or nothing. Research in political science shows that even democracies can be ranked [37]: some are complete and other partial.

## Which capabilities should be included?

Two different views have dominated this debate. Marta Nussbaum [24] suggested a list of dimensions with claims to be universally valid (length of life; health; bodily integrity; senses, imagination and thought; emotions; practical reasons; affiliations; concern for other species; play; and control over one´s environment).

Sen never suggested any definite list of capabilities, because he was not willing to anticipate the processes and decisions that are necessary for each context [38]. We agree, and feel that the choice of capabilities must be tailor-made for a particular setting, although framed by universal principles.

We believe Sen and Nussbaum developed their visions of the capability approach mainly from the perspective of low-income countries. In high-income democracies like Sweden, it is possible to distinguish different clusters of capabilities: those already being equally distributed (e.g. formal political freedom); those offering opportunities to those in need or who have ambitions (e.g. healthcare and free schooling); those meeting vague distributional obligations (e.g. housing); and those being quite far from societal decision-making (e.g. emotions and leisure). Yet societies are seldom neutral, neither in the proportion of the population that votes or finishes education, nor in the distribution of income or pleasure. This is most clearly expressed regarding employment. A high employment rate (including for women) is a pillar in the welfare state [34].

One important point of departure in our research project is to develop a measure useful for public decision-making, and this gives rise to some considerations:

A. Which capabilities are most appropriate for public decision-making?

B. Which capabilities vary mostly between individuals in Sweden today?

Regarding A, we think there are two different angles to consider. One angle is normative: many people would like to have a private sphere in which public interventions are not desired. The other angle is whether public interventions can achieve a certain goal. Love is important in life, but it is not clear if public intervention would be a good idea from a normative point of view, or even effective in terms of benefiting people who feel unloved. Correspondingly, regarding B, formal political rights and free access to nature are not good candidates in Sweden, because a large majority of inhabitants have these capabilities to a level that does not hinder a flourishing life.

Incidentally, the national parliament recently initiated an investigation of possible measures of quality of life in Sweden. The mission was assigned to a professor in sociology, Robert Eriksson, whose expert report suggest capabilities as the best measure of wellbeing in Sweden. The report [39] suggested ten essential capabilities for wellbeing in Sweden: balance of time; economic resources; mental and physical health; political resources; knowledge and skills; sound living environment; employment/commitment; social relations; feeling of security; and safe housing. This was starting point for our purpose, but the next step was to organize a Delphi panel to select particularly relevant capabilities. The panel and its choices are reported in the next section.

## The Delphi panel and the choice of capabilities

### Procedure

Methods for the selection of capabilities have varied. Both expert-led and participatory approaches have been suggested. Nussbaum's famous list is not strictly empirically based.

Researchers in the UK have tried participatory approaches and used qualitative interviews, focus groups and postal surveys [17, 40, 41]. We suggest a further variant, and have worked together with fair minded people in a Delphi process [42] to achieve consensus on a specific question or topic through collaborative decision-making. The process takes place in several rounds in an iterative approach, and information from previous steps is communicated anonymously to participants through a central coordinator or facilitator [43]. The Delphi method is of particular use in the evaluation and assessment of areas in which it is generally difficult to reach a common view or agreement [44]. Most importantly, the Delphi method is useful for exposing priorities of personal values or social goals on a collective basis [44].

The participants of the Delphi panel were selected through a nomination process that was based on the idea of 'fair-minded people', a concept introduced by Norman Daniels [42]. 'Fair-minded people' refers to individuals who agree on mutually justifiable terms of cooperation, and "who want to play by agreed-upon rules and prefer rules that are designed to bring out the best in that game".

The participants were also assumed to have experience of the subject and be willing to dedicate time to the Delphi process (cf. Ogbeifun et al. [43]). We thus invited 22 national (mainly non-profit) organizations (Table 1) to nominate two delegates each (one woman and one man) to take part in our panel. The selection of organizations was influenced by the United Nations Agenda for Sustainable Development for 2030 [45]; we tried to cover individual goals such as no poverty and good health, in the sense that at least one of the organizations had their main activities in these areas. To cover the goal of decent work and economic growth, we invited unions and one enterprise organization, despite these hardly meeting the non-profit criterion.

Some of the goals are more individual and others more collective. Access to clean water, sustainable electricity and combat of climate change are in our terminology collective. Deteriorations affect more or less the whole population and improvements benefit all. Furthermore, there are equity goals, in line with the thinking in our model. Individual goals are:

1. No poverty

2. No hunger

3. Good health and wellbeing

4. Quality education

8. Decent working conditions and economic growth

16. Peace, justice, and strong institutions

In order to introduce the panel members to the research project, and what a Delphi panel means and what their role was, they were invited to a start-up day in Stockholm, where representatives of nine organizations took part. Our research team presented the project, the Delphi process, and the role of the panel. The members in the panel agreed to act like fair-minded people, using the following definition:

"Fair-minded people are not selfish but try to act in the best interest of others; they listen to and think about others' arguments, and they take time to reflect before they take a standpoint."

The procedure in a Delphi panel was explained, and the members were even encouraged to have a dialogue with their partners before answering.

In the first round, the task for the panel was to rank ten capabilities suggested in a Swedish public investigation in 2015, where 1 was the most important and 10 the least important. The ten capabilities were: financial situation, health, education and skills, occupation, social

**Table 1. Actors from the Swedish civil society who participated in the Delphi panel.**

| Name of organisation | Related SDG goals | Answered | Participated |
|---|---|---|---|
| Amnesty | 16: Peace and Justice Strong Institutions | No | |
| Children's Rights in Society | 1: No Poverty | Yes | Yes |
| | 5: Gender Equality | | |
| Crime Victim Support Sweden | 16: Peace and Justice Strong Institutions | Yes | Yes |
| The National Association for Cancer Patients | 1: No Poverty | No | |
| | 3: Good Health and Well-being | | |
| Disability Human Rights | 1: No Poverty | Yes | Yes |
| | 3: Good Health and Well-being | | |
| | 8: Decent Work and Economic Growth | | |
| | 16: Peace and Justice Strong Institutions | | |
| The Swedish Network of Refugee Support Groups | 1: No Poverty | No | |
| | 16: Peace and Justice Strong Institutions | | |
| Islamic Association in Sweden | 1: No Poverty | No | |
| | 16: Peace and Justice Strong Institutions | | |
| National Organization for Pensioners | 1: No Poverty | No | |
| | 3: Good Health and Well-being | | |
| | 16: Peace and Justice Strong Institutions | | |
| Swedish Pensioners' Association | 1: No Poverty | No | |
| | 3: Good Health and Well-being | | |
| | 16: Peace and Justice Strong Institutions | | |
| Comrade Association of Former Criminals | 4: Quality Education | No | |
| | 16: Peace and Justice Strong Institutions | | |
| Fryshuset Global (supporting young people) | 16: Peace and Justice Strong Institutions | No | |
| The Swedish Federation for Lesbian, Gay, Bisexual, Transgender, Queer, and Intersex Rights | 3: Good Health and Well-being | Yes | Yes |
| | 16: Peace and Justice Strong Institutions | | |
| Red Cross | 1: No Poverty | Yes | Yes |
| | 2: Zero Hunger | | |
| Save the Children | 1: No Poverty | No | |
| | 2: Zero Hunger | | |
| | 4: Quality Education | | |
| The Swedish Trade Union Confederation | 8: Decent Work and Economic Growth | Yes | Yes |
| The National Organization for White Collar Workers | 8: Decent Work and Economic Growth | Yes | No |
| The Swedish Confederation of Professional Associations | 8: Decent Work and Economic Growth | Yes | Yes |
| Confederation of Swedish Enterprise | 8: Decent Work and Economic Growth | No | |
| Salvation Army | 1: No Poverty | Yes | Yes |
| | 2: Zero Hunger | | |
| The Swedish Council for Information on Alcohol and Other Drugs | 3: Good Health and Well-being | No | |
| | 16: Peace and Justice Strong Institutions | | |

SDG, Sustainable Development Goals.

relations, residence, security, time balance, local community, and political rights. The panel was invited to suggest further capabilities if they felt anything important was lacking. They were also encouraged to explain their rankings in free text.

To evaluate the first-round rankings, we calculated the median and mean. We also counted the number of times a certain capability was ranked among the top five. The last method requires no interval properties.

### Outcome

After the first round (16 out of 18 answered), the panel strongly agree to include:

- Health, social relations, financial situation, and residence.

  They strongly agree to exclude:

- Education and skills, time balance and local community.

The members gave different rankings for occupation, security and political rights. At the start of the second round (12 answered), the results from the first round were presented, and the panel was given the following new task: to rank occupation, security and political rights (i.e. the capabilities where there were diverging rankings).

The research team suggested that financial resources and residence should be merged into one capability because the two are closely related. People having financial resources are able to find a proper residence. We asked the panel if they would support this change and they agreed.

Finally, we asked the members to confirm the inclusion of health, social relations, and financial situation including residence.

The second round did not come closer to a consensus regarding occupation, security and political rights. They still got almost the same levels of support. However, a large majority of the members approved the merger of financial situation and residence. They also confirmed the inclusion of health, social relations and financial situation (including residence).

We decided not to initiate a third round because the assessment was that this would not increase consensus among the participants. Instead, the number of capabilities included was increased to a total of six. The capabilities selected by the panel were: health, social relations, financial situation including residence, occupation, security, and political rights.

## Setting sufficient capability thresholds in Sweden

Mitchell et al. [19] define a sufficient threshold "as the level of capability at or above which a person's level of capability wellbeing is no longer a concern for policy". Instead, public policies should focus on helping all to reach this level, and in particular to lift those worst-off. However, in the general welfare state those beyond the threshold are still entitled to benefits such as free schooling or subsidized health care. If they were exempted from these benefits many may drop below the sufficient level.

How to set concrete thresholds that are useful in policymaking is not a well-researched topic. We think William's 'fair innings' approach can bring insights [31]. Williams investigated lifetime QALYs in different social groups to suggest a fair inning. Similarly, we thus aim to measure the distribution of capabilities in the Swedish population to set a threshold.

Additionally, when considering both efficiency and equity, some people are willing to trade-off efficiency in exchange for a more equitable distribution, as in the landmark paper by Atkinson [46]. We considered inequality aversion in the population, which may vary between life spheres.

### Distribution of capabilities in the population

With the goal of setting a fair innings threshold for the phrasing of the CALY-SWE question-naire, we conducted a web survey in June 2020 among 1,500 participants from the general Swedish population using a commercial web panel [47]. The aims were: i) to assess the distri-bution of the capabilities held in the population; ii) to explore the impact of different wordings in the capability statements; and iii) to investigate the degree of inequality aversion regarding income, education, and health in the form of life expectancy. The Swedish Ethical Review Authority approved the study with an advisory statement (Dnr 2019–02848) and participants

consented to participate electronically. We excluded answers from participants that stated an age below 18 (ethical concerns) and over 99 (data quality concerns). Detailed methods, participants' characteristics, limitations, and Swedish question phrasings including English translations, are available in S1 File.

The design of the study is summarized in Table 2. Three samples of about 500 persons which were similar regarding sex, age and education were drawn (A, B, and C). We deemed a sample size of 500 as sufficient where with a 5% significance level, a two sample test for proportions detects a 0.1 proportion difference with at least 80% power. The versions differed in how the extent of the capability was described. The A version used the wording of "always" having the capability; the B version used the wording "almost always"; and the C version used the wording "mostly". The B and C versions also had an additional descriptive clause that specified the extent of "almost always" and "mostly". Participants needed to select either "completely agree", "partially agree", or "do not agree at all" for all statements.

Aim i) was to explore the distribution of capabilities. In our model, those who "completely agree" have already sufficient capability and those who "do not agree at all" are those "worst-off". The intention behind "worst-off" is to identify a small group that has the poorest living condition. For the validity and practical usability of CALY-SWE, the size of this proportion should be limited. If the proportion would be, say 50%, the term "worst-off" would lose its meaning. A correct size for the proportion of "worst-off" may not be determinable, but a reasonable proportion may be five to ten percent.

To be useful in policymaking, the highest capability level ("completely agree") should not have been achieved by a large majority but nevertheless not be unattainable for a majority. If too many achieved the highest level, it may be difficult to evaluate reforms that affect the whole distribution, make cross-sectional comparisons, or follow population trends over time.

Aim ii) was to investigate whether the difference in wording between A, B and C influenced the answer pattern. Due to the higher threshold for a "completely agree" answer implied by the stricter wording of A ("always") compared to B ("almost always") and C ("mostly"), we expected an increased proportion of "completely agree" answers for C compared to B and for B compared to A. If this expectation would not get support, we assumed that this way of fine-tuning the phrasing was not meaningful.

The most common answer among participants was "completely agree" followed by "partially agree" for all capabilities except for political resources, where most participants answered "partially agree" followed by "completely agree" (Fig 1).

**Table 2. Design of distribution study.**

| Version and phrasing of capability extent (sample size) | Answer options |
|---|---|
| A: "Always" | a. Completely agree |
| (n = 497) | |
| | b. Partially agree |
| | c. Do not agree at all |
| B: "Almost always" | a. Completely agree |
| (n = 503) | |
| | b. Partially agree |
| | c. Do not agree at all |
| C: "Mostly" | a. Completely agree |
| (n = 505) | |
| | b. Partially agree |
| | c. Do not agree at all |

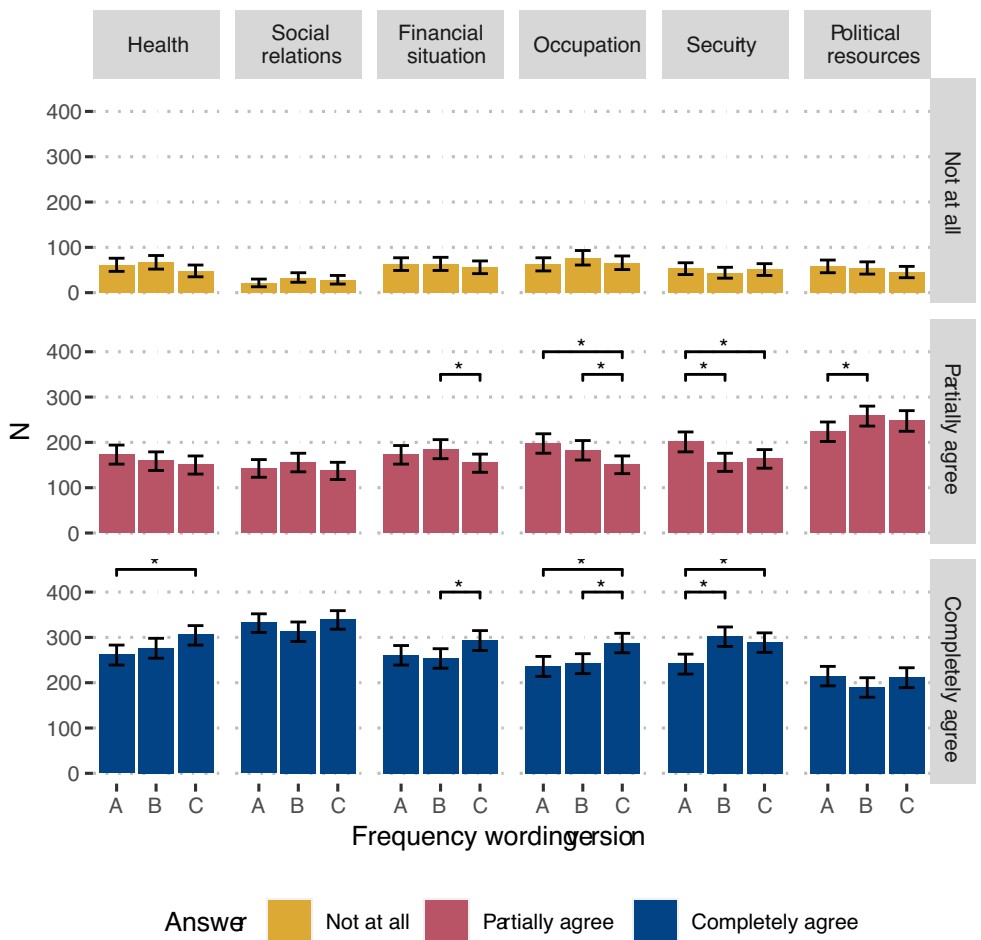

**Fig 1. Distribution of answer alternatives in different wording versions, with bootstrapped 95% confidence intervals.** Significant differences with p < 0.05 in a z-test for difference in proportions are marked with*.

The proportion of "not at all" answers, corresponding to the "worst-off", was stable at around 10 per cent between versions A, B, and C, and the change of frequency in the statements from "always" (A) to "mostly" (C) had no impact. "Completely agree", the part of the population that has reached the threshold of sufficient capabilities, was around 50 per cent. The expected increase of "Completely agree" answers for the versions with "almost always" and "mostly" wordings notably occurred for health, financial situation & housing, occupation and security, where either the proportion for the C compared to the B, C compared to A, or A compared to B was significantly higher. The answer patterns for social relations and political rights did not correspond with the expected increase of "Completely agree" answers in the B and C versions.

## Inequality aversion

We asked participants about the current Swedish distribution in life expectancy, income, and education. Participants could state that the difference was too large or was acceptable, and for income also if it was too small. Detailed phrasings are available in S1 File.

There was a large difference in inequality aversion for income compared to education and life expectancy as a proxy for health. About two thirds of participants judged the differences

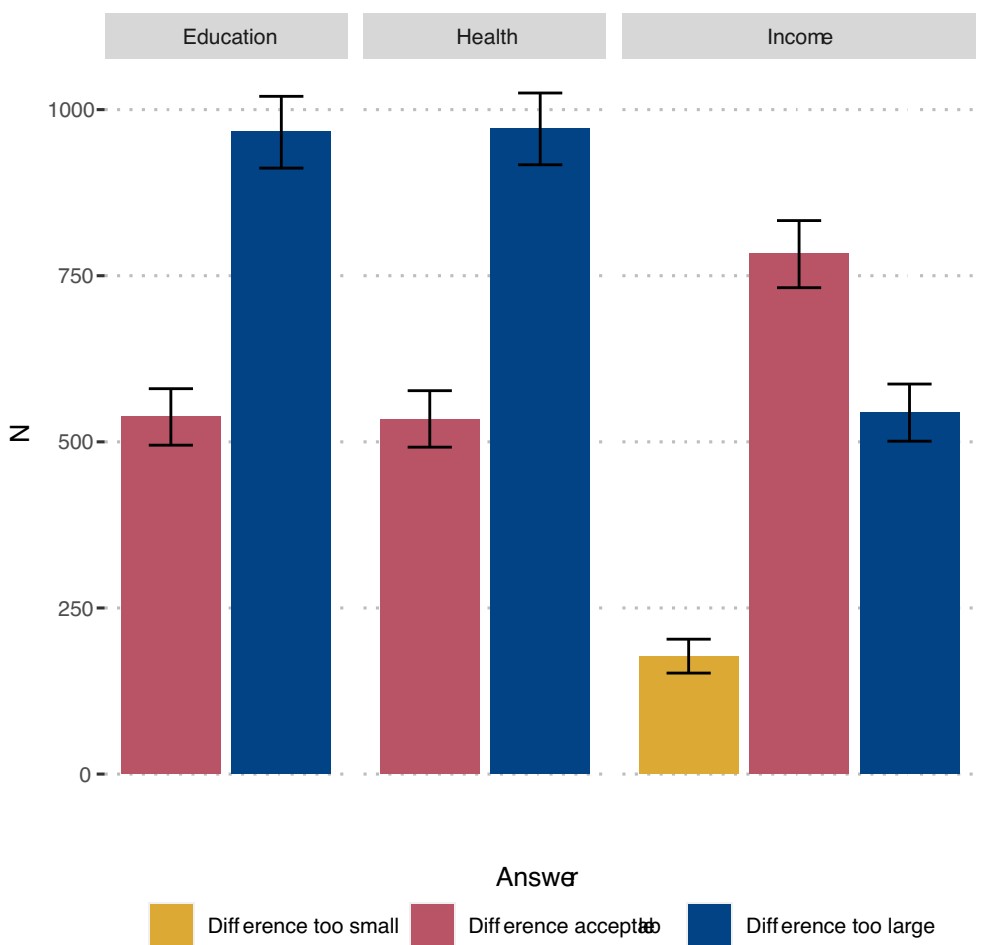

**Fig 2. Distribution of answers in inequality aversion questions with 95% confidence intervals.** Answer alternative 'Difference too small' not available for education and health.

for education and health to be too large, in contrast to income, where only one third found the difference to be too large (Fig 2).

In the next section, 'Capability Levels: Phrasing the statements', we use these findings to phrase the CALY-SWE statements.

## Capability levels: Phrasing the statements

In this section, we suggest how to phrase the level of a capability sufficient for a flourishing life. We discuss in light of the theory of public goods whether it is meaningful for all capabilities to set an explicit threshold. We try to consider all relevant arguments of which we are aware. In particular, we use input from the Delphi process and the study of distribution of capabilities in the population sample. Alongside these two investigations, we consider Swedish laws and policies, and normative ideas in general, which we judge to be of relevance.

**Health:** Based on the input from the Delphi panel, we consider physical and mental health as one of the most (maybe the most) important capabilities. The population survey indicated a pronounced inequality aversion. The threshold should facilitate a high and equitable level of health for everybody. The Healthcare Act [48] and other policy documents [49] point in the

same direction. However, "always" having the health to do what one wants may be unrealistic and certainly extremely demanding of resources, and thus complicates the realization of better health for those worst-off, as well as the realization of other capabilities. Therefore, we think the formulation "almost always" is the most reasonable "fair innings".

**Financial situation and housing:** Compared to health and social relations, a high degree of equality in financial resources may be less relevant for policymaking, as differences in incomes and accommodation are more acceptable. The population survey indicated a mild inequality aversion. From a global perspective, the living standard in Sweden is very high. Considering climate change and sustainable development, it is desirable to reduce the average consumption of goods and services in this country. Yet there are poor groups, especially among immigrants, single parents, and some segments of pensioners. As was reported earlier, around 10 per cent in our study did not agree at all in any of the versions. The proportion of "agree completely", in the "mostly" (C) version compared to the "always" (A) and "almost always" (B) versions was higher, indicating an effect of the phrasing difference. We chose "always" for accommodation and "mostly" for financial situation as the capability threshold—i.e. the fair innings.

**Occupation:** This capability has the highest degree of reciprocity. Work is the foundation of the welfare state and a widespread view is that all should contribute according to their capacity. However, high unemployment rates are a big problem, particularly among young people and people with poor education. From the worst-off angle, certainly unemployment dominates. As for financial situation and accommodation, the wording made a difference, in that "agree partially" proportions were lower and "agree completely" proportion were higher in the "mostly" (C) version compared to the "always" (A) and "almost always" (B) versions. Summing up the arguments above, we think "mostly" is a reasonable societal ambition—i.e. a fair inning. The first priority in policy in this area must be to increase opportunities for the unemployed to get a job, or for those wanting a meaningful activity—e.g. mainly retired or long-term ill people.

Security and political rights have in common that these capabilities can mainly be increased by the provision of public or social goods (or services)—that is, a good) that is both non-excludable and non-rivalrous [50, 51], in that individuals cannot be excluded from use or could benefit from it without paying for it, and where use by one individual does not reduce availability to others, or the good can be used simultaneously by more than one person. Security is a classic example. More police officers in a certain area benefit all living there. The same goes for measures aimed at preventing crime. Similarly, the execution of political rights does not reduce any other person's opportunity to execute his or her rights.

Social relations are a kind of 'micro social good' [52] in the sense that improved relations for one individual correspond to improved relations for at least one another individual.

The answer patterns regarding political and civil rights and social relations indicated that the wording frequencies may were not well understood, and proportions did not differ significantly between versions. For security, the differences in answers between the versions corresponded to the theoretical expectation, with a higher proportion of "completely agree" in the "almost always" (B) and "mostly" (C) versions compared to the "always" (A) version. However, tolerating some degree of unsafety related to violence and crime may collide with article 3 of the human rights declaration that guarantees security of person [53] or the Swedish constitution articles 1 and 2 that proclaim equal treatment and bodily integrity of individuals [54].

Therefore, for political and civil rights, security and social relations, we rejected all the three tested versions. Instead, our phrasing does not include any term for how frequently the capability is available.

In the case of political and civil rights, we use plural wording to underline the nature of political decisions. These are typically laws or policies which treat all individuals the same.

"We count everyone as one, no one for more than one," as Bentham and Mill argued regarding aggregation of utility [55, 56].

The finalized questionnaire is available in S2 File.

## Discussion

The main result of this research is a capability-based model for the economic evaluation of public health and social policies in Sweden. The intention is to support public decision-making and thereby contribute to an increase in wellbeing in the society. The ethical foundation is to consider both equity and efficiency goals, giving particular weight to the interest of the worst-off, trying to make a compromise which can be defended by relevant and convincing arguments.

We judge our proposal to be a logical extension of the cost per QALY approach which is now common in healthcare and public health all over the world. As Cookson [13] argues, QALYs is an application of Sen's capability approach. The frontier between health-related quality of life and global quality of life is fluent. Well-known QALY measures include, for instance, ability to work (EQ5D) [14, 57] and social relations [58]. Thus, there is a continuous transition from pure health measures to general wellbeing measures, rather than a distinct difference.

This paper presents a complete questionnaire, which itself can be used for evaluation and population survey. We recently used the questionnaire for an evaluation of the COVID-19 pandemic [59]. We are also planning to develop a weight tariff using time trade-off and discrete choice evaluation questions.

In Table 3, we have tried to compare Nussbaum's [20] list of capabilities and the Swedish list. A first observation is the language style. Nussbaum's language is more poetic and the Swedish prosaic. However, when we try to overcome this difference, there are many similarities.

All capabilities in the Swedish list are in Nussbaum's list, but not vice versa. Regarding material capabilities, the commitments in the Swedish list are more far-reaching—e.g. Nussbaum wrote "having the right to seek employment" [24]. The corresponding wording in the Swedish list puts a threshold on the function, the right (and duty) to have a job. The same is true for residence, the right to have a permanent residence.

We think that differences in phrasing are adaptations to the different settings—in our case, the Swedish welfare state. All already have the right to seek employment and thus we need to be more specific and committed. Our statement is rather about a function—whether most of the time in previous years, they have held an occupation they are quite happy with. We also argue there is a moral difference between being able to work but not taking a job in real life,

**Table 3. Relation of CALY-SWE dimensions to Nussbaum's list of central capabilities.**

|  | Central capability | Description |
|---|---|---|
| Health | 1,2 | A life of normal length and good health. |
| Financial situation | 10B | Being able to hold property. Having the right to seek employment. |
| Social relations | 5 | Being able to have attachments to things and people outside ourselves. Being able to live with and towards others. |
| Political | 10A | Being able to participate effectively in political choices. |
| Security | 3 | To be secure against violent assault. |
| Occupation | 10B | Being in work, being able to work as human being. Play for retired people. |

and having a job. All share the benefits of the welfare state, such as free education and healthcare, and should feel an imperative to share the burden of producing goods and services and contributing to the welfare state.

Another capability that clearly needs to be adapted to different settings is political rights. We can find many democracies classified as satisfactory (Sweden is one), where people formally hold reasonable rights [37]. However, the extent to which people execute these rights varies considerably, and one important factor may be in the political system [60, 61].

We have chosen to label our measure CALY-SWE, and the reason is, of course, to stress the dependency on setting. As discussed above, the result of our investigation is a quite general set of capabilities. We thus believe they would be valid in neighbouring countries such as Norway, Denmark, and Finland, and probably also in other countries in Northwest Europe. On the other hand, these countries have a much higher population density, which perhaps influences the choice of capabilities. We hope that future research will bring more clarity to this question.

## Is there a conflict between prioritizing the worst-off and the welfare state's general policies?

A symbol of the general policies in Sweden has been child benefit for all, even those families with the highest incomes. Advocates of this policy acknowledge that the material gains for families with high incomes are negligible. Instead, the main reason for the policy is that it contributes to trust and fairness [62] if all taxpayers are also generally eligible for all the benefits in the welfare state. In our model, child benefits to families beyond the threshold in the capability "financial situation" would not contribute to further capability or to more CALYs. However, if the advocates of the welfare state are right, our "political rights" capability would be sensitive to the child benefits scheme. It has been argued [62] that high levels of trust are a condition for the welfare state, perhaps even a consequence of it.

Healthcare is distributed according to need, and almost free of cost at the time of consumption. Healthcare is mainly financed by taxes, and resources are redistributed over the lifespan. There is even a redistribution between the healthy and the ill in the population. As stated above, healthcare in the welfare state already applies a worst-off criterion.

## Experts, fair-minded people, and population surveys

The development of CALY-SWE has utilized the knowledge and perspectives of experts, citizens and a hybrid of those groups—fair-minded-people. The purpose of this mix was to get advice from advocates of objective wellbeing as well as subjective wellbeing. The list of ten capabilities we used as the point of departure was clearly a work of experts. However, fairminded people were not experts, but rather wise laymen given the opportunity to reflect on the topic and take part in a dialogue over some weeks. They were instructed to act in the public interest and not to base their positions mainly on their personal preferences. They should try to imagine the interests of others and try their best to satisfy the interest of the public majority.

We did consider two alternative approaches to the Delphi panel we undertook. One was to use a panel design, but to let political parties nominate the members. This design would link very well to a tradition where the parties nominate, for example, jury members. The reason for us not to choose this approach was a fear that current political disagreements would colour the work of the panel. Furthermore, we also believe that our panel of fair-minded people has deeper insight into disadvantaged people's living conditions. The other approach we considered was a population survey. However, we saw problems with a survey among researchers and doctoral students in health-related medicine [63]. We could not reasonably expect the respondents to use more than about 30 minutes to answer our questions, and that meant an

unreasonable imbalance between the task and the opportunity given for solving it. Secondly, population survey respondents may be guided by their personal preferences, which would lead to a purely subjective model. Thirdly, participants in a one-time cross-sectional survey would not be able to respond to the input of other survey participants, and thus a consensus as in the Delphi panel would not have been possible.

The result from the Panel can be compared with a previous pilot study. The aim of the study was to investigate whether it was possible to rank capabilities included in the Swedish parliament report [39]. The participants were researchers or PhD-students, mostly in public health. The main finding was that most of the respondents managed to rank the capabilities included in the set. The results suggest that participants deemed health to be most important, followed by social relations and financial situation. Knowledge, occupation, time, security, political resources, housing, and living environment were ranked lower, and the exact order depends on the metric used to synthesise the individual rankings.

To form a Delphi panel with fair-minded people is a rather novel approach. The usual Delphi panels consist of experts with formal merits in relation to the panel topic. Our approach will be evaluated in a coming study.

CALYs, like QALYs, require weighting of each set of capabilities. Our intention is to estimate the weights in population surveys. In previous surveys [64], the respondents have assigned higher importance to the step from "not agree" to "partly agree" higher than the step from "partly agree" to "completely agree". We interpret this pattern as a particular concern about the worst-off.

## CALY-SWE compared to other proposals

There are several other instruments based on the capability approach. We provide an incomplete overview of the main approaches and how they compare to CALY-SWE in Table 4. A comprehensive review was recently published [15] that provides a more complete and exhaustive overview of the field.

**Table 4. Comparison of capability-based instruments.**

| | ICECAP-A | ICECAP-O | ICECAP-SCM | OxCAP, OCAP-18, and OxCAP-MH | ASCOT | CALY-SWE |
|---|---|---|---|---|---|---|
| Choice of dimensions | Qualitative interviews | Qualitative interviews | Qualitative interviews | Indicators mapped to Nussbaum's list | Building on previous measures, qualitative interviews, literature review, Delphi process | Delphi process among civil society representatives |
| Purpose | Economic evaluation in general population | Economic evaluation of health and social care interventions in elderly aged 65+ | Economic evaluations for individuals at end-of-life | General population (OxCAP-MH: Mental illness) | General population | 1 Economic evaluation 2 Mapping living conditions over time, between areas |
| Dimensions | Stability, attachment, autonomy, achievement, enjoyment | Attachment, security, role, enjoyment, control | Choice, love and affection, physical suffering, emotional suffering, dignity, being supported, preparation | Life, bodily health, bodily integrity, senses, imagination and thought, emotions, affiliation, other species, play, control over one's environment | Personal cleanliness and comfort, accommodation cleanliness and comfort, food and drink, safety, social participation and involvement, occupation, control over daily life, dignity | Financial situation and housing, health, social relations, occupation, security, and political and civil rights |
| Valuation method | Best–worst scaling | Best–worst scaling | N/A | N/A | Best–worst scaling | Time trade-off and discrete choice |
| Key references | [16, 65] | [40, 66] | [67, 68] | [17, 69, 70] | [18] | [1] |

The ICEpop CAPability measure for adults (ICECAP-A) is a measure of capability wellbeing for economic evaluation aimed at the general adult population [2]. It includes the dimensions of stability, attachment, autonomy, achievement, and enjoyment, selected through qualitative interviews with English informants. The developed questionnaire with four levels per dimension was scored using best–worst scaling [65]. ICECAP-A builds on the ICEpop CAPability measure for older people (ICECAP-O), also featuring five dimensions of attachment, security, role, enjoyment and control, with four levels that were selected by analysing qualitative data [40] and evaluated using best–worst scaling. The ICECAP-Supportive Care Measure (ICECAP-SCM) focuses on quality of life at the end of life [67, 68].

Another family of capability-based measures includes the Oxford CAPability (OxCAP), Oxford CAPability 18 (OCAP-18), and the Oxford CAPability Mental Health (OxCAP-MH) instruments [8–10]. The selection of dimensions was also based on qualitative data and guided by Martha Nussbaum's list of capabilities [24].

The adult social care outcome toolkit (ASCOT) was developed to offer a tool comparable to QALYs for social care. It features the eight dimensions of personal cleanliness and comfort, accommodation cleanliness and comfort, food and drink, safety, social participation and involvement, occupation, control over daily life, and dignity, each on four levels. This selection was based on previous research and the perspective of researchers and users of social care. Tariffs were estimated using best–worst scaling [11]. ASCOT carer is a related version for measuring social care-related quality of life from the perspective of caregivers [71]. In comparison, CALY-SWE focus on the Swedish context, policy relevance, and a paternalistic selection of capabilities in a Delphi process involving fair-minded people.

Thresholds for policy relevant capability levels have also been investigated and operationalized using the ICECAP measure in Mitchell et al. [19] and Goranitis et al. [20]. In contrast to our approach, the sufficient capability level was exploratively set to the second and third highest of four levels without empirical investigation. Kinghorn [72] organised deliberative workshops for citizens, and in total took 62 persons part in 8 workshops. One task was to assess the level of sufficient capabilities, defined by ICACAP-A. The level chosen in the workshops was 3,3,3,3,3 with the best possible state being 4,4,4,4,4.

In our model the threshold is defined by specifying the extend which a capability is available (always, almost always, or mostly). A person who chose the answer option "fully agree" has reached the "sufficient" level. Similar to ICECAP-A, the thresholds for the different capabilities may vary. The sufficient level in CALY-SWE level is based on a synthesis of relevant sources such as policies and laws, the Delphi-panel, present distribution, and inequity aversion in the population.

CALY-SWE also explicitly includes special consideration of those "worst-off" (not agree at all). Priority one in policy is to lift those worst-off, priority two to support those who have not reached a sufficient level, but do not belong to those worst-off (partly agree).

All the attempts described above to establish a sufficient level may be quite immature and explorative, and more research is needed in this area. The level for the CALY-SWE measure may, for instance, change over time due to societal progress.

## Conclusions

The Delphi panel investigation resulted in the capabilities health, social relations, financial situation including residence, occupation, security and political rights to be included in the CALY-SWE measure. We also defined a sufficient level for living a flourishing life for each capability dimension, based on a "fair innings" approach, which allowed us to finalize the CALY- SWE Questionnaire.

## Supporting information

**S1 File. Survey.** Screenshots, phrasing with English translation, methods, and results of the web survey and phrasing in Swedish with English translations.
(DOCX)

**S2 File. CALY SWE questionnaire.** Finalized CALY SWE phrasings.
(DOCX)

## Acknowledgments

We thank Sebastian Östlund from the Department of Historical, Philosophical and Religious studies at Umeå University for giving valuable comments and suggestions. We also thank the Delphi panel participants for their engagement and fair-minded spirit.

## Author Contributions

**Conceptualization:** Kaspar Walter Meili, Anna Månsdotter, Jan Hjelte, Lars Lindholm.

**Data curation:** Kaspar Walter Meili.

**Formal analysis:** Kaspar Walter Meili, Anna Månsdotter, Linda Richter Sundberg, Jan Hjelte, Lars Lindholm.

**Funding acquisition:** Anna Månsdotter, Lars Lindholm.

**Investigation:** Kaspar Walter Meili, Anna Månsdotter, Linda Richter Sundberg, Jan Hjelte, Lars Lindholm.

**Methodology:** Kaspar Walter Meili, Anna Månsdotter, Linda Richter Sundberg, Jan Hjelte, Lars Lindholm.

**Project administration:** Anna Månsdotter, Lars Lindholm.

**Supervision:** Anna Månsdotter, Jan Hjelte, Lars Lindholm.

**Validation:** Kaspar Walter Meili, Anna Månsdotter, Linda Richter Sundberg, Jan Hjelte, Lars Lindholm.

**Visualization:** Kaspar Walter Meili, Lars Lindholm.

**Writing – original draft:** Kaspar Walter Meili, Lars Lindholm.

**Writing – review & editing:** Kaspar Walter Meili, Anna Månsdotter, Linda Richter Sundberg, Jan Hjelte, Lars Lindholm.

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
