## [Decision Letter · Decision Letter 0]

3 Mar 2021

PONE-D-21-04330

An Initiative to Develop Capability-Adjusted Life Years (CALYs) in Sweden: Selecting Capabilities with a Delphi Panel and Developing the Questionnaire

PLOS ONE

Dear Dr. Meili,

Thank you for submitting your manuscript to PLOS ONE. After careful consideration, we feel that it has merit but does not fully meet PLOS ONE’s publication criteria as it currently stands. Therefore, we invite you to submit a revised version of the manuscript that addresses the points raised during the review process.

Please ensure that your revision satisfies PLOS ONE’s publication criteria and not, for example, on novelty or perceived impact.

The reviewer makes three sets of comments which you need to address but accepts this would warrant publication - hence the decision.

The second set:

I just do not see any value in the second survey. The three samples are small (500) and there are different samples in each. The whole objective and methodology just seem totally bizarre to me. I do not see the contribution of this, and I do not see the value in publishing it.

suggests that either you reduce this section and mention it only briefly or you persuade the reviewer it is valuable. The reviewer does not say what appears strange and so the more obvious route would be to downsize the material.

The reviewer comments:

Nor, I'm afraid, do I see the logic of using the survey data to inform the wording of the attribute. Surely this is just perpetuating existing inequalities? Discussion around the third objective felt very subjective and speculative.

So albeit a discussion, you should find literature or sharper arguments that support your view. Also in the limitations section you should mention the possibility of perpetuating inequalities.

If these things are done, the technical questions should be addressed. Please include these comments in your cover letter of reply.

We look forward to receiving your revised manuscript.

Kind regards,

Paul Anand

Academic Editor

PLOS ONE

Journal Requirements:

2. In the Methods section, please provide additional information on participant recrtuiemnt for the online web survey. In particular please  describe any inclusion and exclusion criteria's used. And please provide a justification for the sample size used in your study, including any relevant power calculations (if applicable).

Finally, please provide additional details regarding participant consent. In the ethics statement in the Methods and online submission information, please ensure that you have specified (1) whether consent was suitably informed and (2) what type you obtained (for instance, written or verbal). If your study included minors under age 18, state whether you obtained consent from parents or guardians. If the need for consent was waived by the ethics committee, please include this information.

Reviewers' comments:

Reviewer's Responses to Questions

**Comments to the Author**

1. Is the manuscript technically sound, and do the data support the conclusions?

Reviewer #1: No

2. Has the statistical analysis been performed appropriately and rigorously? 

Reviewer #1: Yes

3. Have the authors made all data underlying the findings in their manuscript fully available?

Reviewer #1: Yes

4. Is the manuscript presented in an intelligible fashion and written in standard English?

Reviewer #1: Yes

5. Review Comments to the Author

Reviewer #1: The paper seeks to achieve 3 objectives: 1) identify the most vital capabilities for individuals in Sweden, 2) to define a sufficient level of each identified capability to lead a flourishing life, and to 3) develop a complete questionnaire for the measurement of the identified capabilities.

The first objective is achieved through taking a long list of capabilities from an existing report and using a Delphi method to arrive at a shorter list. One immediate question is, that if the original list by Eriksson was deemed to be a list of 10 "essential" capabilities, why were all ten not included? Why was there a need to reduce the number of capabilities and refine the list (through merging two of the capabilities)? There does not appear to have been any scope for participants to add in anything that they considered to be missing from the original list of 10.

"Fair minded" individuals were recruited, largely from NGO type organisations. I wonder if there was a danger that the organisations which accepted the invitation to participate approached the task with a particular organisational agenda?

The organisations included children's charities, but no reference is made to whether the questionnaire is intended for adult or child completion. If it is only for adults, then what was it the childrens charities brought to the Delphi task?

I do have serious reservations about:

1) The appropriateness of calling the set of capabilities a "Capability Adjusted Life Years". The QALY is a broad approach whereby information on health is combined with information on life expectancy. Various questionnaires can be used to assess health. We don't refer to these questionnaires as QALYs. This paper is not the first to suggest CALYs, it has been done by Goranitis et al and by Mitchell et al previously, in relation to ICECAP (incidentally, these authors are not referenced, and hence key literature is missed). So, the approach of CALYs is not novel and a CALY could, in theory be calculated from any capability questionnaire. So I do not feel it is appropriate to refer to one set of capabilities by the term associated with a broader approach.

2) The wording of the questionnaire and whether it can actually be considered to assess capability at all - I would say it assesses functioning. I am not clear how the authors went from the original set of capabilities (as described by Eriksson) to the wording used in their survey? The phrasing of some capabilities is quite complicated, awkward and restrictive - for example, there will be lots of junior researchers who get paid enough to afford safe and acceptable accommodation, it will not be a lack of money that prevents them from securing long term accommodation, but instead short term contracts and possibly the need to move to take up new jobs. It is difficult for people to respond to questions which combine several different concepts, and I'm not sure how reliably the information could be interpreted.

Whilst there are a few additional details, and some additional justification that I would like to see provided in relation to the Delphi task, I am prepared to accept the process as been sufficiently sound as to warrant publication. However, I have serious concerns about the extent to which the second two objectives have been met.

I just do not see any value in the second survey. The three samples are small (500) and there are different samples in each. The whole objective and methodology just seem totally bizarre to me. I do not see the contribution of this, and I do not see the value in publishing it.

Nor, I'm afraid, do I see the logic of using the survey data to inform the wording of the attribute. Surely this is just perpetuating existing inequalities? Discussion around the third objective felt very subjective and speculative.

I had a few other, presentational issues:

- Is the rationale for merging capability and QALYs entirely pragmatic? I don't see any conceptual justification?

- Page 11, line 209: The argument here is confusing. First, I don't understand what is meant by "satisfaction of capabilities". Then, I fail to see how increasing income would not enhance a person's capability? I don't think a very good understanding of sufficiency is demonstrated, and again, references to sufficiency are missing.

- Should we be allocating more resources to those in the poorest health? Does this not depend on their ability to respond to treatment, and the cost and cost-effectiveness of that treatment?

Table 4 makes reference to TTO/DCE, but I can't see that this has been discussed in the paper.

6. PLOS authors have the option to publish the peer review history of their article (what does this mean?). If published, this will include your full peer review and any attached files.

Reviewer #1: No

---

## [Author Response · Author response to Decision Letter 0]

25 Apr 2021

Dear Reviewer and Editor

We highly appreciate your efforts to review our manuscript and the valuable comments. We carefully considered your feedback and revised our manuscript accordingly. 

We specifically shortened the section on the methods and results of the empirical survey considerably by shifting a substantial part to the supplementary. We also revised the section to improve clarity and structure, revised Figure 1 to contain the information in Figure 2 in a more understandable format and removed Figure 2. 

We also agree with the reviewers reasoning regarding the name. In 2017 [1] we suggested CALYs as a proper name for measures inspired by Sen’s capability approach and the QALY-approach. All “QALYs” a measure of health-related quality of life and a time dimension in common. The scale for health-related quality of life is anchored between “perfect health” and a death. Drummond also mentions an additional a value premisses: ”Also it should be remembered that the cost-per-QALY ranking does embody a kind of equality, in that a QALY is considered to be worth the same to every individual” [2].

However, in this family of “QALYs”, different methods are used to construct health related quality of life measures. Well-known are EQ5D, SF36/SF6 and HUI, and established methods for estimating weights are TTO and standard gamble. Despite these differences, we label the construct(s) QALY.

Our thinking regarding CALYs follow the same lines. As for QALYs, different methods will be developed, and the content in the measure will certainly varies between countries. Even different methods to assign weights to states will be used. We think capability measures in general can be transformed to CALY, if the states can be properly located on a scale from 0 to 1. Finally, we think equity should be considered in CALYs, for example as in the “sufficient capabilities” approach [3], or our fair-innings model. But other methods might exists or will be developed for trading of equity and efficiency when using CALYs as a measure for resource allocation.

In our paper from 2017, we refer to and describe the content in Mitchell et al. [3]. Of course should Mitchell et al. (and Goranitis et al. [4]) be in the present reference list, and we apologize our carelessness. 

Please find our point-by-point response below. 

Yours sincerely

The authors

Original editor comments

Reply: We renamed the figure files to start with a capital letter.

2. In the Methods section, please provide additional information on participant recrtuiemnt for the online web survey. In particular please  describe any inclusion and exclusion criteria's used. And please provide a justification for the sample size used in your study, including any relevant power calculations (if applicable).Finally, please provide additional details regarding participant consent. In the ethics statement in the Methods and online submission information, please ensure that you have specified (1) whether consent was suitably informed and (2) what type you obtained (for instance, written or verbal). If your study included minors under age 18, state whether you obtained consent from parents or guardians. If the need for consent was waived by the ethics committee, please include this information.

Reply: We included the following sentences in the section Setting sufficient capability thresholds in Sweden->Distribution of capabilities in the population: Regarding inclusion exclusion criteria “We excluded answers form participants that stated an age below 18 (ethical concerns) and over 99 (data quality concerns).”and regarding power analysis “We deemed a sample size of 500 as sufficient where with a 5% significance level, a two sample test for proportions detects a 0.1 proportion difference with at least 80% power.”We also mention exclusion criteria and power analysis in S1 Survey, under the heading Material and methods. The only exclusion criteria we applied were age < 18 or age > 99. In the same section we mention that consent was obtained using the online survey. Note that beyond age we did not apply any inclusion and exclusion criteria.

Where:

Lines 383-384Lines 388-390S1 supplement, Material and methods

Reply: Thank you for highlighting this. We share PLOS one’s philosophy open access to underlying data. The data is currently under review at the Swedish National Data to be published according to the FAIR principles. We will notify you as soon as the data is published.

In-house check April 21

1) In the Methods section, please provide additional information on participant recrtuiemnt for the online web survey. In particular please describe any inclusion and exclusion criteria's used. And please provide a justification for the sample size used in your study, including any relevant power calculations (if applicable).Finally, please provide additional details regarding participant consent. In the ethics statement in the Methods and online submission information, please ensure that you have specified (1) whether consent was suitably informed and (2) what type you obtained (for instance, written or verbal). If your study included minors under age 18, state whether you obtained consent from parents or guardians. If the need for consent was waived by the ethics committee, please include this information.

Reply: Thank you for your request. We added “and participants consented electronically to the sentence “The Swedish Ethical Review Authority approved the study with an advisory statement (Dnr 2019-02848) and participants consented to participate electronically.” Also see comment 2. above regarding sample size and inclusion exclusion criteria.

Where:

Lines 382-383

2) Thank you for including your ethics statement on the online submission form: "The Swedish Ethical Review Authority approved the study with an advisory statement (Dnr 2019-02848)." To help ensure that the wording of your manuscript is suitable for publication, would you please also add this statement at the beginning of the Methods section of your manuscript file.

Reply: Thank you for pointing out this issue. We included a modified version of the sentence in the section Setting sufficient capability thresholds in Sweden->Distribution of capabilities in the population, after the aims. See also 1).

Where:

Lines 382-383

Reviewer

Set 1

1) The appropriateness of calling the set of capabilities a "Capability Adjusted Life Years". The QALY is a broad approach whereby information on health is combined with information on life expectancy. Various questionnaires can be used to assess health. We don't refer to these questionnaires as QALYs. This paper is not the first to suggest CALYs, it has been done by Goranitis et al and by Mitchell et al previously, in relation to ICECAP (incidentally, these authors are not referenced, and hence key literature is missed). So, the approach of CALYs is not novel and a CALY could, in theory be calculated from any capability questionnaire. So I do not feel it is appropriate to refer to one set of capabilities by the term associated with a broader approach.

Reply: We agree with your reasoning. And we propose ‘CALY-SWE’ as a proper name. This is what we adopted in the manuscript. In the revision we consistently use CALY-SWE where applicable, but retain ‘CALY’ where we talk about capability-adjusted life years in general. To give appropriate credit to the previous applications and conceptualizations of CALYs we mention the authors you list and some others we changed the sentence leading up to the aims in the Introduction; from “However, to our best knowledge, the amalgamation of the QALY approach and a set of capabilities covering vital aspects of a good life other than health, is novel.” to “While capability-adjusted life years have been suggested and implemented before [3–6], our approach includes equity considerations from the start and is focused on an amalgamation of the QALY approach and a set of capabilities covering vital aspects of a good life other than health.” The references consist of Goranitis et al., Mitchell et al., the 2008 report by Lorgelly et al, and and the 2017 ICECAP application by Walters et al.

Where:

Through the manuscriptLine 155

Set 2

2) The wording of the questionnaire and whether it can actually be considered to assess capability at all - I would say it assesses functioning. I am not clear how the authors went from the original set of capabilities (as described by Eriksson) to the wording used in their survey? The phrasing of some capabilities is quite complicated, awkward and restrictive - for example, there will be lots of junior researchers who get paid enough to afford safe and acceptable accommodation, it will not be a lack of money that prevents them from securing long term accommodation, but instead short term contracts and possibly the need to move to take up new jobs. It is difficult for people to respond to questions which combine several different concepts, and I'm not sure how reliably the information could be interpreted.

Reply: Thank you for your comment. One of the strongest initiatives from the Delpi-panel was to raise the social problems we have in Sweden due to housing shortage. Some people are literally homeless, many more cramped. Prices have skyrocket, and in the big cities only high-income earners can afford a reasonable housing standard. The Panel also defended the view that a welfare state is committed to implement policies which make it possible to get permanent housing within a reasonable waiting-time, and to a reasonable cost, for all who wants. This should be considered as a capability. If anyone prefer to live with temporary contracts they are free to do so. To give university students temporary contracts during the years they study is a practical arrangement, and would not be interpreted as a limitation of the capability. However, all the young that are forced to stay with parents because they cannot afford own housing suffer a limitation.To facilitate the interpretation of this capability we plan to have a question about housing in the background section.In the Discussion, we explain why occupation is formulated as a function. One reason is the construction of the welfare state. All who share the benefits of the welfare state (free health care, education etc,) should also share the burdens.

Where:

Line 536

I just do not see any value in the second survey. The three samples are small (500) and there are different samples in each. The whole objective and methodology just seem totally bizarre to me. I do not see the contribution of this, and I do not see the value in publishing it.

Reply: The logic of our approach is heavily inspired by Alan Williams “Fair Innings” [7]. He describes expected QALYs during lifetime in different social groups, and against this empirical pattern he suggests a fair inning to be, say 70 QALYs. If he had set the fair inning to be 80 years, none of the groups had reached it and if he had set it equal to sixty all would have gotten it. A fair inning in some African country would of course be different from the fair inning in UK, so information about present distribution in the particular setting is helpful.Our idea is simply to follow in the foot-steeps of Williams. We want to prioritize those worst-off (also referring to Rawls theory of justice), which are those who “not at all” agree to the stated capability. This threshold needs to take the present distribution of a capability into account, exactly as Williams did. If the whole population would be below the threshold, we cannot identify those worst-off. “All” can of course not be worse off. In our data, we found that around 10% were worst-off, and this proportion remained stable between the three samples.With a sample size of 500, the power to detect a proportion difference of 0.1 between two survey versions, using a two sample test for proportions, is at least 80%, which we deemed adequate for our purposes. We aimed for independent samples because we believe that asking the different capability wordings for each capability in sequence would lead to invalid results as participants might change in how they answer between the wording versions. The whole section has been shortened, and partly rewritten. We have changed the heading to “Setting sufficient capabilities thresholds in Sweden” to better describe the content. The section has also got a new introduction to clarify the logic.

Where:

Lines 363-433

Nor, I'm afraid, do I see the logic of using the survey data to inform the wording of the attribute. Surely this is just perpetuating existing inequalities? Discussion around the third objective felt very subjective and speculative.

Reply: Mitchell et al. [3] discuss how sufficient capabilities can be defined in ICECAP-O. This instrument has four levels for each capability. Full capability (level 4), a lot of capability (level 3), a little capability (level 2) and no capability (level 1). The threshold is set by choosing a level (i.e. 3) and the threshold need not to be constant over all capabilities. It can vary depending on the real availability of capabilities in a certain context.In our model the threshold is defined by specifying the extend which a capability is available (always, almost always etc). A person who chose the answer option “fully agree” has reached the “sufficient” level. In similarity with ICECAP-O varies the thresholds for the different capabilities.Neither ICECAP-O nor our method are “objective”. Rather, to set a threshold is a value-based decision.We have tried to be transparent, and describe the different aspects which have been taken into account.How to set concrete threshold for sufficient capabilities so that they are useful in policy-making is not a well-researched topic. We see it thus as a first generation method that we and others try to develop, and they will certainly be refined in the future.

Where:

Lines 436-503

Other, representational issues

- Is the rationale for merging capability and QALYs entirely pragmatic? I don't see any conceptual justification?

Reply: We think there are good reasons to define “severity” of a disease as the loss of quality of life times the duration of the state. Seasickness is terrible in the moment, but since you completely recover when you disembark the ferry, it’s not judged to be a serious condition. We think this time dimension is crucial when we judge all capabilities. It matters whether you lack money for a month or suffer poverty for decades. Thus, the inclusion of time is a deliberate, value driven decision.Additionally, extending the QALY concept to incorporate capabilities seems logical from a welfare theory point of view, as we outline in the section from Introduction-> From welfarism to extra-welfarism and capabilities section:In parallel to the development of the capability approach and extra-welfarism based on the shortcomings of welfarism, we see CALYs as an application of the capability approach in line with extra-welfarist ideas to solving problems of resource allocation. There are indeed conceptual points of friction with the capability approach as well, such as the stricter descriptive system that allows for complete orderings, but we think that our concept goes beyond pure pragmatism.

Where:

Lines 103-158

- Page 11, line 209: The argument here is confusing. First, I don't understand what is meant by "satisfaction of capabilities". Then, I fail to see how increasing income would not enhance a person's capability? I don't think a very good understanding of sufficiency is demonstrated, and again, references to sufficiency are missing.

Reply: Thank you for highlighting this unclarity related to “satisfaction” which may be connotated with “preference”. We changed the wording in this sentence to read “does not increase capabilities”. We also changed other uses of “capability satisfaction” in the manuscript. The question if our model perpetuates inequalities is central. In fact, our model is constructed to promote equality. Indeed, increased income enhances capability. However, moving into making priorities between conflicting societal goals, e.g. “the equity-efficiency trade-offs”, one idea could be to put a threshold (say annually 100 000 Euro) that is good enough for giving reasonable opportunities for a flourishing life. Mitchell et al (2016) defines the threshold “as the level of capability at or above which a person’s level of capability wellbeing is no longer a concern for policy”.On the other end of the income distribution we have people earning only 10 000 Euro annually, certainly not enough for a flourishing life. In our model an increase from 10 000 to say 20 000 would yield increased capabilities in the societal calculus. For instance and individual could change from “Not agree” to “Partially agree” to having the capability. Contrary, an increase from 100 000 to 110 000 would not yield anything in the societal calculus, because already 100 000 leads to the answer “Completely agree” to having the capability. Therefore only polices improving the conditions for those below the threshold would be able to improve the level of capabilities. In particular, policies benefiting those “worst-off” would yield new levels of capabilities. This is a parallel to the fair innings idea by Alan Williams (1997). In resource allocation in health care, priority should be given to those who did not get their fair inning. Our model extends this principal view to all capabilities.

Where:

Line 214, 476, 560

- Should we be allocating more resources to those in the poorest health? Does this not depend on their ability to to treatment, and the cost and cost-effectiveness of that treatment?

Reply: We agree that resource allocation to people with poor health should be dependent on the cost-effectiveness of the treatment (which ideally would factor in the access to the treatment). Furthermore, the mainstream normative assumption in cost-effectiveness analysis is health maximization, that is to purely rank treatment according to the cost-effectiveness. This idea has its roots in utilitarianism, and has been repeatedly challenged also within health economics, such as by Wagstaff [8] and Williams [7]. Both argue for abandoning pure health maximization and instead make a trade-off between efficiency and equity.We can add that pure health maximization is not a reasonable goal in policy-making. In Sweden, and many other countries, equity considerations play a prominent role in public decision-making, and our “model” (CALY-SWE) is intended to facilitate decision-making. Also Sen’s own writing supports equity considerations and criticize utility or health maximization [9,10]

Table 4 makes reference to TTO/DCE, but I can't see that this has been discussed in the paper.

Reply: Thank you for the comment. We added a sentence that briefly mentions that we plan on using TTO/DCE questions: “We are also planning to develop a weight tariff using time trade -off and discrete choice evaluation questions.”

Where:

Line 521

---

## [Decision Letter · Decision Letter 1]

12 Aug 2021

PONE-D-21-04330R1

An Initiative to Develop Capability-Adjusted Life Years in Sweden (CALY-SWE): Selecting Capabilities with a Delphi Panel and Developing the Questionnaire

PLOS ONE

Dear Dr. Meili,

Thank you for submitting your manuscript to PLOS ONE. After careful consideration, we feel that it has merit but does not fully meet PLOS ONE’s publication criteria as it currently stands. Therefore, we invite you to submit a revised version of the manuscript that addresses the points raised during the review process.

We look forward to receiving your revised manuscript.

Kind regards,

Rabia Hussain

Academic Editor

PLOS ONE

Journal Requirements:

Reviewers' comments:

Reviewer's Responses to Questions

**Comments to the Author**

1. If the authors have adequately addressed your comments raised in a previous round of review and you feel that this manuscript is now acceptable for publication, you may indicate that here to bypass the “Comments to the Author” section, enter your conflict of interest statement in the “Confidential to Editor” section, and submit your "Accept" recommendation.

Reviewer #1: (No Response)

2. Is the manuscript technically sound, and do the data support the conclusions?

Reviewer #1: Partly

3. Has the statistical analysis been performed appropriately and rigorously? 

Reviewer #1: Yes

4. Have the authors made all data underlying the findings in their manuscript fully available?

Reviewer #1: Yes

5. Is the manuscript presented in an intelligible fashion and written in standard English?

Reviewer #1: Yes

6. Review Comments to the Author

Reviewer #1: The manuscript is improved and I think I understand the motivation for the online survey now - the authors seem to set the sufficient level as the level that most people already achieve in the small population surveys. I still don't understand why the same survey was not sent to all 1,500 respondents, with a request for them to indicate whether they can achieve the capability somewhat, mostly or always?? Surely this would have made more sense than seeing how many people from a smaller sub-sample strongly agree that they can achieve the capability somewhat, how many people from a separate and equally small subset strongly agree that they can achieve the capability mostly, etc?? It just seems like an unnecessarily complicated way of getting the information, which breaks the sample down into three small and independent sub-samples.

Also, I think it could still be clearer that the objective was to match the sufficient level to the existing distribution of achievement. And I STILL think it needs to be acknowledged that this does NOT reflect societal values about what constitutes a good life, but instead risks perpetuating poor performance of social policies or existing inequalities. Where is the incentive for social policies to drive improvements in quality of life if all they need to do is perpetuate existing achievement? Will the sufficient level need to be up-dated over time?

There are a few things that are misleading and need to be changed:

Page 9: It is wrong to suggest that ICECAP measures only focus on health, in fact they don't explicitly include health at all. So the work reported here is NOT novel in the sense of moving beyond health outcomes.

Page 10 - I STILL do not believe that the work contributes a sufficient level for a FLOURISHING LIFE (see above) - at no point have the authors asked anyone or considered what constitutes a good life. Instead the work establishes (loosely!!) a rough indication of current achievement on the attributes - the majority of people may have perfectly miserable lives - this is never established or considered.

Lines 396 to 401 on page 23 - I don't understand what is being said here.

Discussion - Kinghorn HAS used empirical methods to establish a sufficient level of capability well-being for ICECAP-A, but their work (published in Social Science & Medicine) is not acknowledged.

7. PLOS authors have the option to publish the peer review history of their article (what does this mean?). If published, this will include your full peer review and any attached files.

Reviewer #1: No

---

## [Author Response · Author response to Decision Letter 1]

9 Sep 2021

Dear Reviewer and Editor

We would like to extend our appreciation for your efforts in reviewing our manuscript and for providing valuable comments. 

We think that the comments and remarks helped us to further improve the manuscript and would like to thank the reviewer. 

Please consider our point-by-point response below on the ensuing pages for the changes in the manuscript. 

Yours sincerely

Kaspar Walter Meili 

Anna Månsdotter

Linda Richter Sundberg 

Jan Hjelte

Lars Lindholm

 

Editor comment:

“Please review your reference list to ensure that it is complete and correct. If you have cited papers that have been retracted, please include the rationale for doing so in the manuscript text, or remove these references and replace them with relevant current references. Any changes to the reference list should be mentioned in the rebuttal letter that accompanies your revised manuscript. If you need to cite a retracted article, indicate the article’s retracted status in the References list and also include a citation and full reference for the retraction notice.”

Answer: 

We went through all the items in the bibliography and failed to find a rejected publication. We however detected a wrongly cited item: Bibliography item [46] (Fenner F et al., Family and generic names for viruses approved by the International Committee on Taxonomy of Viruses) was cited by mistake on line 368, the correct citation would have been item [31] (Williams A, Intergenerational Equity: An Exploration of the ‘Fair Innings’ Argument). We corrected the issue, the citation now is on line 372. 

Reviewer comments:

Major 1:

“The manuscript is improved and I think I understand the motivation for the online survey now - the authors seem to set the sufficient level as the level that most people already achieve in the small population surveys. I still don't understand why the same survey was not sent to all 1,500 respondents, with a request for them to indicate whether they can achieve the capability somewhat, mostly or always?? Surely this would have made more sense than seeing how many people from a smaller sub-sample strongly agree that they can achieve the capability somewhat, how many people from a separate and equally small subset strongly agree that they can achieve the capability mostly, etc?? It just seems like an unnecessarily complicated way of getting the information, which breaks the sample down into three small and independent sub-samples.”

Answer: 

Thank you for the comment and the relevant remark. The intention with not sending the same survey to all the participants was to enable the 3 subsamples that are similar and representative of the Swedish population to answer the statements individually, independently from each other. We wanted to use wording in the statements that is similar to the final version of the questionnaire. With the suggested study design this would have not been possible because 

-1) (if all participants get the 3 versions) the answers to the different statements would not be independent of each other, possibly biased, 

or 

-2) (if as the reviewer suggested, a question phrasing in the style of “do you achieve the capability X?”- somewhat, -mostly, -always, would have been used) a phrasing quite dissimilar to the final questionnaire and different answer options would have been necessary to keep a respectably level of face validity and the results would not have the same legitimacy to guide the choice of phrasing for the statements. 

Major 2:

“Also, I think it could still be clearer that the objective was to match the sufficient level to the existing distribution of achievement. And I STILL think it needs to be acknowledged that this does NOT reflect societal values about what constitutes a good life, but instead risks perpetuating poor performance of social policies or existing inequalities. Where is the incentive for social policies to drive improvements in quality of life if all they need to do is perpetuate existing achievement? Will the sufficient level need to be up-dated over time?”

Answer: 

Thank you. We agree that the section that outlines the goals of the survey is not enough clearly written, and the part between lines 387 and 401 is rewritten to increase clarity. Among other things, we think a short table (Table 2) can be used to explain the design of the study. We cannot exclude that the sufficient level and phrasing requires to be updated. Indeed, adaptions may be necessary in the future to reflect progress or changed societal values. 

The revised pars are on lines 367 to 370, 391 to 392 and 401 to 420. 

Minor 1: 

“Page 9: It is wrong to suggest that ICECAP measures only focus on health, in fact they don't explicitly include health at all. So the work reported here is NOT novel in the sense of moving beyond health outcomes.”

Answer: 

Thank you for your comment, what we write on page 9 is : “While capability-adjusted life years have been suggested and implemented before [1–4], our approach includes equity considerations from the start and is focused on an amalgamation of the QALY approach and a set of capabilities covering vital aspects of a good life relevant other than health.”

For instance, we use time-trade off to establish weights as some QALY approaches do.

We know and even write about measures (ICACAP and other) which have moved beyond health. We did by no means intend to suggest that ICECAP does focus health and apologise. We therefore changed the wording to (on line 158): 

“… vital aspects of a good life relevant for the Swedish policy context.”

Minor 2: 

“ I STILL do not believe that the work contributes a sufficient level for a FLOURISHING LIFE (see above) - at no point have the authors asked anyone or considered what constitutes a good life. Instead the work establishes (loosely!!) a rough indication of current achievement on the attributes - the majority of people may have perfectly miserable lives - this is never established or considered.”

Answer:

What we argue in the paper, in similarity with other capability approaches, is to give people sufficient capabilities for a flourishing life. Lack of capabilities should not hinder people form living the lives they want. However, sufficient capabilities are not a guarantee for flourishing life. Even if all capabilities met high threshold, there would certainly be people dissatisfied with their lives and living conditions. 

The choice of capabilities in our model has been a process in several steps: the governmental investigation, a survey among public health researchers and finally the Delphi-panel with members from different not-for-profit organisations.

When it comes to the level of a capability sufficient for a flourishing life, we try to consider all relevant arguments of which we are aware. In particular, we use input from the Delphi process and the study of distribution of capabilities in the population sample. Alongside these two investigations, we consider Swedish laws and policies, and normative ideas in general, which we judge to be of relevance.

For example, about health we write (on line 463-471 in the revised manuscript):

“Health: Based on the input from the Delphi panel, we consider physical and mental health as one of the most (maybe the most) important capabilities. The population survey indicated a pronounced inequality aversion. The threshold should facilitate a high and equitable level of health for everybody. The Healthcare Act [49] and other policy documents [50] point in the same direction. However, “always” having the health to do what one wants may be unrealistic and certainly extremely demanding of resources, and thus complicates the realization of better health for those worst-off, as well as the realization of other capabilities. Therefore, we think the formulation “almost always” is the most reasonable “fair innings”. “

The other thresholds are set using a similar synthesizing reasoning.

Minor 3:

Lines 396 to 401 on page 23 - I don't understand what is being said here.

Answer: 

We reworded the section to increase clarity from

“Due to the higher threshold implied by the wording of A (“always”) compared to B (“almost always”) and C (“mostly”), we expected an increased proportion of “completely agree” on C compared to B and on B compared to A. Our intent was to select the wording version with a distribution pattern that fitted normative considerations well. No difference between the versions was an argument for adapting the A version with the shortest and least complex formulation.”

to (now on lines 416 to 420)

“Due to the higher threshold for a “completely agree” answer implied by the stricter wording of A (“always”) compared to B (“almost always”) and C (“mostly”), we expected an increased proportion of “completely agree” answers for C compared to B and for B compared to A. If this expectation would not get support, we assumed that this way of finetuning the phrasing was not meaningful.”

Please also refer to Major comment 2.

Minor 4: 

Discussion -Kinghorn HAS used empirical methods to establish a sufficient level of capability well-being for ICECAP-A, but their work (published in Social Science & Medicine) is not acknowledged. 

Answer: 

Thank you, we include now the 2019 Kinghorn paper now in the discussion on line 658 and relate it to our approach on lines 656 to 673: 

… “Kinghorn [72] organised deliberative workshops for citizens, and in total took 62 persons part in 8 workshops. One task was to assess the level of sufficient capabilities, defined by ICACAP-A. The level chosen in the workshops was 3,3,3,3,3 where the best possible state is 4,4,4,4,4. 

In our model the threshold is defined by specifying the extend which a capability is available (always, almost always, or mostly). A person who chose the answer option “fully agree” has reached the “sufficient” level. Similar to ICECAP-A, the thresholds for the different capabilities may vary. The sufficient level in CALY-SWE level is based on a synthesis of relevant sources such as policies and laws, the Delphi-panel, present distribution, and inequity aversion in the population.

CALY-SWE also explicitly includes special consideration of those “worst-off” (not agree at all). Priority one in policy is to lift those worst-off, priority two to support those who have not reached a sufficient level, but do not belong to those worst-off (partly agree).

All the attempts described above to establish a sufficient level may be quite immature and explorative, and more research is needed in this area. The level for the CALY-SWE measure may, for instance, change over time due to societal progress. “

Other Changes: 

We also would like to add the results from a pilot study (published in this Journal ref). We suggest the inclusion of the following paragraph after line 509: 

“The result from the Panel can be compared with a previous pilot study. The aim of the study was to investigate whether it was possible to rank capabilities included in the Swedish parliament report (35). The participants were researchers or PhD-students, mostly in public health. The main finding was that most of the respondents managed to rank the capabilities included in the set. The results suggest that participants deemed health to be most important, followed by social relations and financial situation. Knowledge, occupation, time, security, political resources, housing, and living environment were ranked lower, and the exact order depends on the metric used to synthesize the individual rankings.”

On line 154 we changed “social care” to “social services” to increase clarity (“Capabilities, as an outcome measure in healthcare, elderly care or social services, have gained increasing attention during the last decade”). 

We also consistently applied the spelling “worst-off” throughout the manuscript (“instead of worst off”).

---

## [Editor Report · Decision Letter 2]

17 Jan 2022

An Initiative to Develop Capability-Adjusted Life Years in Sweden (CALY-SWE): Selecting Capabilities with a Delphi Panel and Developing the Questionnaire

PONE-D-21-04330R2

Dear Dr. Meili,

We’re pleased to inform you that your manuscript has been judged scientifically suitable for publication and will be formally accepted for publication once it meets all outstanding technical requirements.

Kind regards,

Rabia Hussain

Academic Editor

PLOS ONE
---

## [Editor Report · Acceptance letter]

31 Jan 2022

PONE-D-21-04330R2 

An Initiative to Develop Capability-Adjusted Life Years in Sweden (CALY-SWE): Selecting Capabilities with a Delphi Panel and Developing the Questionnaire 

Dear Dr. Meili:

I'm pleased to inform you that your manuscript has been deemed suitable for publication in PLOS ONE. Congratulations! Your manuscript is now with our production department. 

Kind regards, 

on behalf of

Dr. Rabia Hussain 

Academic Editor

PLOS ONE